# Predictors of glycemic control, quality of life and diabetes self-management of patients with diabetes mellitus at a tertiary hospital in Ghana

Kwadwo Faka Gyan[1]*, Enoch Agyenim-Boateng[1], Kojo Awotwi Hutton-Mensah[1], Priscilla Abrafi Opare-Addo[1], Solomon Gyabaah[1], Emmanuel Ofori[2], Osei Yaw Asamoah[1], Mohammed Najeeb Naabo[1], Michael Asiedu Owiredu[1], Elliot Koranteng Tannor[1,3]

1 Directorate of Medicine, Komfo Anokye Teaching Hospital, Kumasi, Ghana, 2 Department of Family Medicine, Dalhousie University, Yarmouth, Nova Scotia, Canada, 3 Department of Medicine, School of Medical Sciences, Kwame Nkrumah University of Science and Technology, Kumasi, Ghana

* kfgonemission@gmail.com

## Abstract

### Background

The burden of diabetes mellitus (DM) in Sub-Saharan Africa is high and continues to increase. Effective DM management focuses on key goals such as glycemic control, prevention of complications and improvement of quality of life (QOL). This study therefore assessed predictors of glycemic control, QOL and diabetes self-management (DSM) of patients with DM in a tertiary hospital in Ghana.

### Methods

We conducted a cross-sectional study involving face-to-face interviews of patients with DM attending clinic using structured questionnaires and validated study instruments as well as review of medical records. A multivariable logistic regression analysis was used to identify independent factors associated with good glycemic control, poor QOL and poor DSM practices.

### Results

The study involved 360 patients with mean age of 62.5 ± 11.6 years and mean FBG of 9.0 ± 4.8 mmol/L, of which 40.8% had FBG < 7 mmol/L. Patients who were not on insulin(aOR 1.82; 95% CI 1.12–2.96, p = 0.017) were more likely to have good glycemic control. Urban residence (aOR 0.24; 95% CI 0.06–0.87, p = 0.030) was protective of having poor QOL. However, poor DSM(aOR 18.30; 95% CI 7.98–44.5, p < 0.001) and recent hospitalization(within the past 3 months) (aOR 4.58; 95% CI 1.58–13.26, p = 0.005) had higher odds of poor QOL. Patients who were divorced(aOR 6.92; 95% CI 1.24–43.20, p = 0.031) had higher odds of poor DSM, while having attended the

**Data availability statement:** All relevant data are within the manuscript and its Supporting Information files.

**Funding:** The author(s) received no specific funding for this work.

**Competing interests:** The authors have declared that no competing interests exist.

clinic for more than 3 years(aOR 0.32; 95% CI 0.13–0.84, p = 0.018) was protective of poor DSM.

## Conclusion

4 out of 10 patients attending diabetes clinic are well controlled. Not being on insulin is independently associated with good glycemic control. Urban residence, DSM and recent hospitalization are associated with QOL while being divorced and duration in clinic predict DSM. Prevention of acute hospitalizations and promotion of good self-management among patients with diabetes can improve their quality of life.

---

## Introduction

Diabetes Mellitus (DM) is a chronic medical condition with a high morbidity and mortality [1]. According to the World Health Organization (WHO) global report on DM, 422 million people were living with DM in 2014, compared to 108 million in 1980 representing an 80.8% rise in the prevalence [2]. The rise in prevalence is notably more substantial in Sub-Saharan Africa (SSA). The number of persons living with DM aged 20–79 years in Africa is projected to increase by 98% from 12.1 million in 2010 to 23.9 million by 2030, compared with a global average increase of 54% over the same period [3]. Among adult Ghanaians, the overall prevalence of DM is 6.46% according to a systematic review and meta-analysis by Asamoah-Boaheng *et al.* in 2019, compared with 0.2% in 1964 [4].

Global disability-adjusted life years (DALYs) of DM increased by 116.7% from 1990 to 2017 and from 31.3 million to 67.9 million [1]. A pivotal goal in clinical practice is to ultimately improve the quality of life (QOL) of patients. Thus QOL assessment is essential to guide the management of chronic medical conditions such as DM [5]. QOL is defined by the WHO as "an individual's perception of their position in life in the context of the culture and value system where they live, and in relation to their goals, expectations, standards, and concerns" [6]. It therefore juxtaposes patients' expectations with their lived experience. Several studies have demonstrated that, patients with DM have lower QOL compared to the general population due to several factors including treatment demands, disease burden and complications with increased hospitalization and mortality [7–9]. According to a meta-analysis by Teli *et al.*, predictors of QOL include age, marital status, gender, monthly income, education, area of residence, religiosity, DM knowledge and self-efficacy, comorbidities, duration of DM, insulin treatment, physical activity, medication adherence, frequent glucose checks and family support [10]. The overall QOL of patients with DM is low in Ghana and Nigeria with higher QOL associated with medication adherence and employment status [11]. Psychosocial factors such as DM distress, depressive symptoms, family support and diabetes self-management (DSM) activities have also been identified to affect well-being and QOL among Ghanaian patients with DM [12].

Since the management of DM is multi-faceted and includes pharmacological interventions as well as patient-centred lifestyle modifications, goals such as glycaemic

control, prevention of acute and chronic complications and enhancement of QOL may be difficult to achieve if a holistic approach to treatment is not embraced [13]. According to several studies, good DSM is associated with good glycaemic control [14–17]. Good DSM has to do with patients taking control of their condition and adhering to the four thematic areas: dietary control, glucose management, physical activity and seeking care from health professionals. Patients with DM however practice different levels of adherence to DSM. The degree of physiological and glycaemic benefits of DSM depends on the fidelity with which it is practiced. Different studies report varying domain-specific and overall mean scores of DSM using various tools. In Ethiopia and Myanmar, good DSM practice among people living with T2DM was low [16,18] whilst it was high in Jordan and Canada [19,20]. Programs to educate, support and empower patients to undertake DSM practices have thus become the focus of attention and are associated with a significant impact on treatment targets [21–24]. The DSM interventions in a meta-analysis have been found to also effectively improve many physiological outcomes, such as blood pressure control, lipid control, and body mass index (BMI) [25]. According to a systematic review by Lamptey *et al.*, structured DSM education is associated with a reduction in glycated haemoglobin especially in sub-Saharan Africa [26]. However, a randomized controlled trial in Ghana comparing usual care only to usual care plus a structured DSM education program failed to show any association with glycaemic control [27]. Traditionally, DSM education interventions in Ghana are under-resourced and unstructured [28]. Health care workers particularly general nurses, doctors, pharmacists and dieticians are involved in providing appropriate and timely DSM education to patients either individually during consultations or as a group during waiting periods at health facilities. There are also trained DM nurses as well as peer educators involved in DSM education interventions in Ghana.

In this study, we sought to assess the glycemic control, QOL and DSM of patients with DM in a tertiary hospital in Ghana. The predictors of good glycemic control, poor QOL and poor DSM were further determined. These predictors when properly identified can be targeted with interventions to improve glycemic control, QOL and DSM of patients with DM in Ghana.

## Materials and methods

### Study design

This was a cross-sectional study involving face-to-face interviews using structured questionnaires and validated study instruments as well as medical record reviews by study physicians.

### Study setting and population

The study was conducted at the DM Clinic of the Komfo Anokye Teaching Hospital (KATH). KATH is a 1200 bed capacity tertiary-level health facility located in Kumasi in the Ashanti region of Ghana. It is the second-largest tertiary centre in Ghana and receives referrals from all over the country. The DM clinic is a specialist-led clinic with over 40 patients in attendance daily. Trained DM nurses educate patients on DSM, albeit unstructured, every morning at the start of the clinic.

### Inclusion criteria

- Patients diagnosed with DM using the American Diabetes Association (ADA) criteria [29]
- Patients attending the DM clinic for at least 6 months
- Patients 18 years or above

### Exclusion criteria

- Diabetic pregnant women or gestational DM

## Sampling procedure

Patients were randomly sampled from the daily attendants at the DM clinic till the calculated sample size was obtained. Patients on arrival at the DM clinic register at the nurses' station and every third patient who met the eligibility criteria was invited for enrolment after explaining the objectives of study and obtaining informed consent. Participants were recruited from 10th May, 2023–25th August, 2023.

## Sample size

A sample size of 360 was determined from Fisher's formula [N = $z^2$ (pq)/$d^2$] at confidence level of 95% (z) (1.96), estimated proportion (p) of participants with poor glycaemic control of 70% from a recent study [30], q = (1- p), margin of error (d) = 0.05 and accounting for a 10% non-response rate [31].

## Data collection and study procedures

A structured questionnaire was administered to each participant to obtain socio-demographic data and medical history.

The World Health Organization Quality of Life-BREF (WHOQOL-BREF) scale, validated in SSA and among patients with diabetes [6,11,32] was used to assess patients' QOL. It comprises 26 items measuring four domains: physical health, psychological health, social relationships and environment, on a 5-point Likert scale ranging from 1 (very dissatisfied) to 5 (very satisfied). It yields total scores for each domain and an overall QOL score. The physical health domain includes activities of daily living, mobility, pain and discomfort, and work capacity. The psychological domain comprise body image and appearance, positive and negative feelings, spirituality and religion. Social support, personal relationships and sexual activity are the facets of social relationships domain while financial resources, safety and security, and access to transport and healthcare comprise the environment domain [32].

Patients' DSM was assessed using the Diabetes Self-Management Questionnaire (DSMQ) [33]. The validated DSMQ scale for full psychometric assessment regarding DM has 16 items and 4 subscales: healthcare patronage (3 items; 3,7,14), glucose management (5 items; 1,4,6,10,12), physical activity (3 items; 8,11,15) and dietary control (4 items; 2,5,9,13) and item 16 is the patient's overall rating of his/her DSM and it is added to the 'Sum Scale' score. In terms of what is regarded as effective diabetes self-care, seven items are formulated positively and the remaining nine negative. The DSMQ has a four-point Likert scale that starts with 0 = does not apply to me, 1 = applies to me to some degree, 2 = applies to me to a considerable degree and 3 = applies to me very much. For individual analysis to be possible, a box is put below each item for ticking if that item is not required in their treatment. During the scoring, all negative word items are reversed such that higher scores indicated more effective self-care. Sums of item scores are calculated to give scale scores and then converted into a scale that ranges from 0 to 10 (raw score/theoretical maximum score *10). In a situation where 'it is not required as part of my treatment' is marked, that item is excluded from the calculation and the theoretical maximum scores reduces accordingly.

Most recent clinical and laboratory data (within 3 months of enrolment) were recorded from patients' medical records.

## Study variables and definitions

Socio-demographic data collected included: age, gender, residence, marital status, educational status, employment status, household income based on the minimum wage, and health insurance status. Clinical data included: type of DM and duration of disease, recent hospitalization (both DM and non-DM-related admissions within the past 3 months), family history of DM, alcohol use, smoking history, presence of co-morbidities, complications, current DM medications and other medications, blood pressure, body mass index (BMI), fasting blood glucose (FBG) and glycated haemoglobin (HbA1c).

Good glycaemic control was defined as HbA1c < 7% [30] or FBG < 7.0 mmol/L [29].

Poor QOL was defined as QOL z-score < −1SD.

Poor DSM was defined as DSM score < −1SD.

## Data analysis

Data collected from the study participants was entered into REDCap (Research Electronic Data Capture) version 15.0.5 ©Vanderbilt University and exported to Stata 17.0 for analysis. The characteristics of study participants were described by using frequencies and percentages for categorical variables; means and standard deviations for normally distributed continuous variables; medians and interquartile ranges for skewed continuous variables. A multivariable logistics regression analysis adjusted for age, gender and residence was fitted to identify factors independently associated with good glycemic control (HbA1c and FBG models), poor QOL and poor DSM. Factors associated with good glycemic control, poor QOL and poor DSM at p-value <0.05 in univariable analysis were selected for inclusion into the multivariable model. Further analysis using the same process outlined was used to determine predictors of poor individual QOL domains and poor DSM domain scores. Statistical analysis was performed using Stata 17.0 and p-values <0.05 were considered statistically significant.

## Ethical considerations

Ethical approval was obtained from Komfo Anokye Teaching Hospital's Institutional Review Board with ethical approval number of KATH IRB/AP/012/23. Written informed consent was obtained from all participants.

## Results

### Socio-demographic characteristics of patients

The study recruited 360 participants with a mean age of 62.5±11.6 years, the majority of them being females 271(75.3%) and residing in urban locations 260(72.2%). It was found that most participants 215(59.7%) were married and those divorced were the fewest 15(4.2%). Most of the participants were unemployed 221(61.5%) and 190(52.8%) had a household income of GH¢541–2700 (USD 46–230) (Table 1).

### Clinical characteristics of patients

The majority of participants had Type 2 DM 342(95.0%) and the median duration of disease was 14 years [IQR:6–20]. Most participants 276(76.7%) had been attendants at the DM clinic for more than three years and 37(10.3%) had been hospitalized in the past 3 months. All participants had FBG available however some participants did not have evaluable HbA1c mainly due to financial constraints. The mean HbA1c (n=215) was 7.8±2.7% and 44.7% had HbA1c<7% while the mean FBG was 9.0±4.8 mmol/L with 40.8% having FBG<7.0mmol/L. There was no statistical difference between the mean FBG of participants with HbA1c and those without HbA1c (p=0.541). The most common comorbidities were hypertension 309(85.8%), dyslipidemia 120(33.3%) and obesity 116(32.2%). We found the most common complications documented to be neuropathy 243(67.5%), retinopathy 181(50.3%), nephropathy 18(5.0%) and stroke 15(4.2%). Metformin was used in 321(89.2%) of patients followed by insulin 153(42.5%), sulphonylureas 142(39.4%) and then thiazolidinediones 94(26.1%). Only 7(1.9%) of patients were on dipeptidyl peptidase-4 inhibitors (DPP4i) with none on sodium-glucose cotransporter-2 inhibitor (SGLT-2i) and glucagon-like peptide-1 (GLP-1) agonist. Majority of patients 291(80.8%) were however on two or more DM medications (Table 1).

### Quality of life and diabetes self-management scores

The overall QOL score for participants using the WHOQOL-BREF scale was 71.0 as a composite of physical health domain 67.9, psychological domain 71.9, social relationships domain 67.6 and environment domain 76.5 (Table 2). The DSMQ sum score for participants was 8.4 with individual domain scores being 7.6 for glucose management, 8.8 for dietary control, 8.0 for physical activity and 9.7 for health-care use (Table 3).

**Table 1.** Characteristics of patients with DM, N = 360.

| Characteristic | Frequency | Percentage (%) |
|---|---|---|
| **Age in years, Mean (SD)** | 62.5 (11.6) | |
| **Gender** | | |
| Male | 89 | 24.7 |
| Female | 271 | 75.3 |
| **Residence** | | |
| Urban | 260 | 72.2 |
| Peri-urban | 82 | 22.8 |
| Rural | 18 | 5.0 |
| **Marital status** | | |
| Single | 30 | 8.3 |
| Married | 215 | 59.7 |
| Separated | 12 | 3.3 |
| Divorced | 15 | 4.2 |
| Widowed | 88 | 24.4 |
| **Educational level** | | |
| Basic | 178 | 49.4 |
| Secondary | 124 | 34.4 |
| Tertiary | 58 | 16.1 |
| **Employment status** | | |
| Employed | 139 | 38.6 |
| Unemployed | 221 | 61.4 |
| **Household income (GH₵)** | | |
| <540 | 125 | 34.7 |
| 541-2700 | 190 | 52.8 |
| >2700 | 24 | 6.7 |
| No response or Unknown | 21 | 5.8 |
| **NHIS or Private insurance** | 357 | 99.2 |
| **Type 2 DM** | 342 | 95.0 |
| **Duration of disease in years, Median (IQR)** | 14 (6–20) | |
| **Duration in clinic** | | |
| 6 months - <1 year | 44 | 12.2 |
| 1-3 years | 40 | 11.1 |
| > 3 years | 276 | 76.7 |
| **Recent hospitalization, <3months** | 37 | 10.3 |
| **Family history of DM** | 236 | 65.6 |
| **Alcohol use** | | |
| Current | 22 | 6.1 |
| Past | 71 | 19.7 |
| Nil | 267 | 74.2 |
| **Smoking history** | | |
| Current | 1 | 0.3 |
| Past | 15 | 4.2 |
| Nil | 344 | 95.6 |
| **Blood Pressure in mmHg** | | |
| SBP, Mean (SD) | 139.6 (24.1) | |
| DBP, Mean (SD) | 79.8 (12.2) | |

*(Continued)*

**Table 1.** (Continued)

| Characteristic | Frequency | Percentage (%) |
|---|---|---|
| **BMI in Kg/m², Mean (SD)** | 27.8 (5.9) | |
| **FBG in mmol/L, Mean (SD)** | 9.0 (4.8) | |
| Good glycemic control (<7) | 147 | 40.8 |
| ***HbA1c in %, Mean (SD) n = 215** | 7.8 (2.7) | |
| Good glycemic control (<7) | 96 | 44.7 |
| **Comorbidities** | | |
| Hypertension | 309 | 85.8 |
| Chronic Kidney Disease | 9 | 2.5 |
| Heart failure | 9 | 2.5 |
| Dyslipidemia | 120 | 33.3 |
| Obesity | 116 | 32.2 |
| None | 10 | 2.8 |
| §Other | 39 | 10.8 |
| **Complications** | | |
| Retinopathy | 181 | 50.3 |
| Nephropathy | 18 | 5.0 |
| Neuropathy | 243 | 67.5 |
| Peripheral arterial disease | 5 | 1.4 |
| Foot ulcer or amputation | 9 | 2.5 |
| Stroke | 15 | 4.2 |
| **DM medications used** | | |
| Metformin | 321 | 89.2 |
| Sulphonylureas | 142 | 39.4 |
| Thiazolidinedione | 94 | 26.1 |
| DPP-4i | 7 | 1.9 |
| SGLT-2i | 0 | 0.0 |
| GLP-1 analogues | 0 | 0.0 |
| Insulin | 153 | 42.5 |
| **Number of DM medications used** | | |
| None or mono- | 69 | 19.2 |
| Dual- or multi- | 291 | 80.8 |

SD: Standard Deviation. GH₵: Ghana Cedis. NHIS: National Health Insurance Scheme. IQR: Interquartile Range. DM: Diabetes Milletus. SBP: Systolic Blood Pressure, DBP: Diastolic Blood Pressure, BMI: Body Mass Index. FBG: Fasting Blood Glucose. HbA1c: Glycated hemoglobin. DPP4i: Dipeptidyl peptidase-4 inhibitors, SGLT-2i: Sodium glucose transporter-2 inhibitor, GLP-1: Glucagon-like peptide-1

*Mean FBG in mmol/L of participants with HbA1c (n = 215) is 9.1 ± 4.9 and of participants without HbA1c (n = 145) is 8.8 ± 4.7 (p = 0.541)

§Cancer (breast ca, lung ca, CLL), Chronic lung disease (Asthma, COPD, lung fibrosis), Neuropsychiatric (Stroke. Parkinson's disease, Schizophrenia, Seizure disorder, Chronic Inflammatory Demyelinating syndrome), Peptic ulcer disease, Cataract, Hyperthyroidism, Arthritis, Benign prostatic hyperplasia.

### Predictors of good glycemic control, poor quality of life and poor diabetes self-management

In a multivariable logistic regression including all significant socio-demographic and clinical variables, patients who were not on insulin (aOR 2.14; 95% CI 1.19–3.88, p = 0.012) were more likely to have good glycemic control compared to those on insulin, when HbA1c (n = 215) was used as a measure of glycemic control (Table 4). Another independent model which used FBG as a measure of glycemic control yielded similar results (aOR 1.82; 95% CI 1.12–2.96, p = 0.017) (Table 5). Urban residence (aOR 0.24; 95% CI 0.06–0.87, p = 0.030) and household income of GH₵

**Table 2. Quality of life scores of patients with diabetes mellitus.**

| Domain score | Transformed scored (0–100), Mean ±SD |
|---|---|
| Physical health | 67.92 ± 20.14 |
| Psychological | 71.88 ± 18.16 |
| Social relationships | 67.64 ± 14.38 |
| Environment | 76.46 ± 15.82 |
| Total score | 71.00 ± 13.23 |

**Table 3. Diabetes self-management scores of patients with diabetes mellitus.**

| Domain Score | Mean ± SD |
|---|---|
| Glucose management | 7.58 ± 2.60 |
| Dietary control | 8.80 ± 1.94 |
| Physical activity | 7.98 ± 3.06 |
| Health-care use | 9.67 ± 1.02 |
| Sum Score(10) | 8.39 ± 1.60 |

541−2700 (USD 46–230) (aOR 0.28; 95% CI 0.13–0.64, p = 0.002) were protective of having poor QOL. However, poor DSM (aOR 18.30; 95% CI 7.98–44.5, p < 0.001) and recent hospitalization (within the past 3 months) (aOR 4.58; 95% CI 1.58–13.26, p = 0.005) had higher odds of poor quality of life (Table 6). Patients who were divorced (aOR 6.92; 95% CI 1.24–43.20, p = 0.031) and those without comorbidities (aOR 5.53; 95% CI 1.18–23.40, p = 0.021) had higher odds of poor DSM, but having attended the clinic for more than 3 years (aOR 0.32; 95% CI 0.13–0.84, p = 0.018) was protective of poor DSM (Table 7).

### Predictors of poor individual QOL domains and poor DSM domain scores

In a multivariable logistic regression including all significant socio-demographic and clinical variables, physical health domain of QOL was predicted by age in years (aOR 1.01; 95% CI 1.05–1.15, p < 0.001), presence of other comorbidities (aOR 6.08; 95% CI 1.76–20.40, p = 0.003); psychological health domain of QOL was predicted by married status (aOR 0.33; 95% CI 0.12–0.92, p = 0.030), being unemployed (aOR 3.28; 95% CI 1.61–7.05, p = 0.002), presence of other comorbidities (aOR 3.84; 95% CI 1.54–9.36, p = 0.003), presence of neuropathy (aOR 3.92; 95% CI 1.93–8.61, p < 0.001); social relationships domain of QOL was predicted by male gender (aOR 2.99; 95% CI 1.63–5.52, p < 0.001), past alcohol use (aOR 2.16; 95% CI 1.15–4.04, p = 0.016), presence of nephropathy (aOR 4.14; 95% CI 1.46–12.90, p = 0.009) and environment domain of QOL by primary level of education (aOR 5.79; 95% CI 1.75–24.50, p < 0.008). Poor DSM was significantly associated with poor QOL in all domains (Table 8).

The DSM domain of glucose management was predicted by urban residence (aOR 0.33; 95% CI 0.12–0.92, p = 0.029); dietary control domain was predicted by being divorced (aOR 14.30; 95% CI 2.44–124.00, p = 0.006), obesity (aOR 7.77; 95% CI 1.05–71.60, p = 0.045), nephropathy (aOR 3.40; 95% CI 1,15–9.42, p = 0.020) and absence of comorbidities (aOR 8.53; 95% CI 2.22–36.7, p = 0.002); physical activity domain by urban residence (aOR 0.11; 95% CI 0.004–0.33, p < 0.001), clinic attendance of more than 3 years (aOR 0.37; 95% CI 0.16–0.92, p = 0.029), presence of nephropathy (aOR 5.36; 95% CI 1.63–17.40, p = 0.005), presence of neuropathy (aOR 3.46; 95% CI 1.62–8.14, p = 0.002) and stroke (aOR 5.20; 95% CI 1.55–16.60, p = 0.006); and healthcare utilization domain by recent hospitalization (aOR 5.54; 95% CI 1.99–14.90, p < 0.001) and presence of peripheral artery disease (aOR 11.60; 95% CI 1.43–76.20, p = 0.010).(Table 9).

**Table 4. Predictors of good glycemic control among patients with DM (HbA1c model) (n = 215).**

| Variables | cOR | 95%CI | p-value | aOR | 95%CI | p-value |
|---|---|---|---|---|---|---|
| **Age in years** | 1.01 | 0.99–1.04 | 0.235 | 1.01 | 0.98–1.03 | 0.499 |
| **Gender, Male** | 1.01 | 0.55–1.85 | 0.975 | 0.99 | 0.53–1.85 | 0.980 |
| **Residence** | | | | | | |
| Rural | 1 | | | 1 | | |
| Peri-urban | 0.48 | 0.13–1.77 | 0.27 | 0.47 | 0.12–1.84 | 0.277 |
| Urban | 0.67 | 0.17–2.64 | 0.564 | 0.79 | 0.19–3.39 | 0.733 |
| **Marital status** | | | | | | |
| Single | 1 | | | – | | |
| Married | 1.72 | 0.62–4.81 | 0.3 | – | | |
| Separated | 3.61 | 0.64–20.32 | 0.145 | – | | |
| Divorced | 1.44 | 0.29–7.10 | 0.651 | – | | |
| Widowed | 2.08 | 0.67–6.39 | 0.203 | – | | |
| **Educational level** | | | | | | |
| Basic | 1 | | | – | | |
| Secondary | 0.91 | 0.50–1.63 | 0.741 | – | | |
| Tertiary | 0.52 | 0.23–1.17 | 0.114 | – | | |
| **Employment status** | | | | | | |
| Employed | 1 | | | – | | |
| Unemployed | 1.19 | 0.68–2.08 | 0.540 | – | | |
| **Household income (GH₵)** | | | | | | |
| <540 | 1 | | | – | | |
| 541-2700 | 0.94 | 0.54–1.65 | 0.831 | – | | |
| >2700 | 0.51 | 0.15–1.80 | 0.296 | – | | |
| **Type 2 DM** | 0.80 | 0.25–2.55 | 0.702 | – | | |
| **Disease duration ≥10years** | 0.89 | 0.51–1.54 | 0.675 | – | | |
| **Duration in clinic** | | | | | | |
| <1year | 1 | | | – | | |
| 1-3years | 1.38 | 0.49–3.95 | 0.543 | – | | |
| >3years | 1.10 | 0.51–2.40 | 0.808 | – | | |
| **Recent hospitalization** | 0.76 | 0.38–1.55 | 0.458 | – | | |
| **Family history of DM** | 1.10 | 0.51–2.38 | 0.811 | – | | |
| **Alcohol use** | | | | | | |
| Nil | 1 | | | – | | |
| Current | 1.20 | 0.46–3.11 | 0.714 | – | | |
| Past | 1.33 | 0.66–2.66 | 0.424 | – | | |
| **Smoking history** | | | | | | |
| Nil | 1 | | | – | | |
| Current | 0 | | | – | | |
| Past | 1.60 | 0.42–6.12 | 0.494 | – | | |
| **Number of DM medications used** | | | | | | |
| None or mono | 1 | | | – | | |
| Dual or multi | 0.54 | 0.28–1.05 | 0.070 | – | | |
| **DM medications used** | | | | | | |
| Insulin | 1 | | | – | | |
| No insulin | 2.10 | 1.21–3.64 | 0.008 | 2.14 | 1.19–3.88 | 0.012 |

*(Continued)*

**Table 4.** (Continued)

| Variables | cOR | 95%CI | p-value | aOR | 95%CI | p-value |
|---|---|---|---|---|---|---|
| **Comorbidities** | | | | | | |
| Hypertension | 1.04 | 0.49–2.22 | 0.912 | – | | |
| Chronic kidney disease | 0.24 | 0.03–2.09 | 0.196 | – | | |
| Heart failure | 1.25 | 0.25–6.32 | 0.79 | – | | |
| Dyslipidemia | 0.62 | 0.35–1.09 | 0.097 | – | | |
| Obesity | 1.24 | 0.08–20.12 | 0.879 | – | | |
| None | 1.25 | 0.25–6.32 | 0.79 | – | | |
| Other comorbidities | 1.75 | 0.70–4.34 | 0.23 | – | | |
| **Complications** | | | | | | |
| Retinopathy | 1.15 | 0.67–1.96 | 0.621 | – | | |
| Nephropathy | 0.49 | 0.09–2.56 | 0.394 | – | | |
| Neuropathy | 1.56 | 0.88–2.76 | 0.125 | | | |
| Peripheral arterial disease | 0.41 | 0.04–3.98 | 0.44 | – | | |
| Foot ulcer or amputation | 1.68 | 0.37–7.70 | 0.503 | – | | |
| Stroke | 0.49 | 0.09–2.56 | 0.394 | – | | |
| **QOL z-score<−1SD** | 1.04 | 0.49–2.19 | 0.920 | – | | |
| **DSM score<−1SD** | 1.38 | 0.67–2.88 | 0.379 | – | | |

HbA1c: Glycated hemoglobin. cOR: Crude Odds Ratio. CI: Confidence Interval. aOR: Adjusted Odds Ratio. GH₵: Ghana Cedis. DM: Diabetes mellitus. QOL: Quality of life. DSM: Diabetes self-management

## Discussion

Our study captures the real world DM management experience at a tertiary hospital in Ghana and identifies not being on insulin to be associated with good glycemic control as compared with patients on insulin. Urban residents and those with good household income had better QOL whereas poor self-management as well as recent hospitalization were associated with poor QOL. Patients who had attended the DM clinic for more than three years had better DSM.

The prevalence of multi-morbidity and complications in our cohort is not surprising as the mean age and duration of DM disease are 63 years and 14 years respectively. A recent systematic review and meta-analysis of 126 peer-reviewed studies that included nearly 15.4 million people showed that more than half the adult population worldwide over 60 years has multiple co-morbidities [34]. Also, longer duration of DM is a known predictor of chronic DM complications [35]. This study showed that 5.0% and 5.6% of patients have nephropathy and established cardiovascular disease, however, none of the study participants were on SGLT-2 inhibitors or GLP-1 agonists.

Good glycemic control was observed in 40.8% of patients using FBG. Glycemic control was also assessed using HbA1c however, some participants did not have evaluable HbA1c with the most significant barrier being cost. In the real world management of patients at the DM clinic in low resource settings, it may be observed that not all patients are able to access HbA1c routinely [36]. In the present study, there was however no statistical difference between the mean FBG of participants with HbA1c and those without HbA1c. Alternatives to assessing glycemic control when HbA1c is not accessible may include using FBG, post prandial glucose and dynamic blood glucose profile based on self-monitoring of blood glucose [36]. Compared to other studies, a cross-sectional multi-centre institution-based study in Western Ethiopia showed that 35.9% of participants had good glycemic control [37] whiles another study at a referral hospital in South-west Ethiopia found that 72.7% of participants had good glycemic control [38]. A multi-centre study in China showed 47.5% of patients had good glycemic control as determined by a fasting plasma glucose less than 7.0 mmol/L [39]. The differences in glycemic control across these studies may be due to the study designs: single-centre versus multi-centre and primary

**Table 5. Predictors of good glycemic control among patients with DM (FBG model).**

| Variables | cOR | 95%CI | p-value | aOR | 95%CI | p-value |
|---|---|---|---|---|---|---|
| **Age in years** | 1.03 | 1.01–1.05 | 0.009 | 1.02 | 1.00–1.04 | 0.088 |
| **Gender, Male** | 1.33 | 0.82–2.16 | 0.248 | 1.15 | 0.68–1.96 | 0.592 |
| **Residence** | | | | | | |
| Rural | 1 | | | 1 | | |
| Peri-urban | 0.64 | 0.23–1.78 | 0.393 | 0.63 | 0.23–1.70 | 0.360 |
| Urban | 0.69 | 0.26–1.79 | 0.444 | 0.70 | 0.24–2.03 | 0.507 |
| **Marital status** | | | | | | |
| Single | 1 | | | – | | |
| Married | 1.81 | 0.79–4.14 | 0.158 | – | | |
| Separated | 1.17 | 0.28–4.88 | 0.833 | – | | |
| Divorced | 0.85 | 0.21–3.93 | 0.816 | – | | |
| Widowed | 1.62 | 0.66–3.93 | 0.290 | – | | |
| **Educational level** | | | | | | |
| Basic | 1 | | | – | | |
| Secondary | 0.72 | 0.45–1.16 | 0.176 | – | | |
| Tertiary | 1.61 | 0.89–2.93 | 0.116 | – | | |
| **Employment status** | | | | | | |
| Employed | 1 | | | – | | |
| Unemployed | 1.04 | 0.67–1.60 | 0.867 | – | | |
| **Household income (GH₵)** | | | | | | |
| <540 | 1 | | | – | | |
| 541-2700 | 1.2 | 0.77–1.87 | 0.411 | – | | |
| >2700 | 1.65 | 0.70–3.94 | 0.255 | – | | |
| **Type 2 DM** | 1.40 | 0.51–3.83 | 0.508 | – | | |
| **Disease duration ≥10years** | 1.11 | 0.71–1.74 | 0.649 | – | | |
| **Duration in clinic** | | | | | | |
| <1 year | 1 | | | – | | |
| 1–3 years | 1.43 | 0.58–3.56 | 0.441 | – | | |
| >3 years | 1.81 | 0.91–3.60 | 0.093 | – | | |
| **Recent hospitalization** | 0.76 | 0.38–1.55 | 0.458 | – | | |
| **Family history of DM** | 0.76 | 0.38–1.55 | 0.458 | – | | |
| **Alcohol use** | | | | | | |
| Nil | 1 | | | 1 | | |
| Current | 0.92 | 0.37–2.28 | 0.864 | 0.86 | 0.33–2.24 | 0.765 |
| Past | 1.76 | 1.04–2.98 | 0.035 | 1.58 | 0.90–2.80 | 0.114 |
| **Smoking history** | | | | | | |
| Nil | 1 | | | – | | |
| Current | 0 | | | – | | |
| Past | 2.27 | 0.79–6.51 | 0.129 | – | | |
| **Number of DM medications used** | | | | | | |
| None or mono | 1 | | | – | | |
| Dual or multi | 0.57 | 0.33–0.96 | 0.034 | 0.68 | 0.39–1.19 | 0.177 |
| **DM medications used** | | | | | | |
| Insulin | 1 | | | – | | |
| No insulin | 2.26 | 1.45–3.51 | <0.001 | 1.82 | 1.12–2.96 | 0.017 |

*(Continued)*

**Table 5.** (Continued)

| Variables | cOR | 95%CI | p-value | aOR | 95%CI | p-value |
|---|---|---|---|---|---|---|
| **Comorbidities** | | | | | | |
| Hypertension | 1.19 | 0.65–2.19 | 0.575 | – | | |
| Chronic kidney disease | 1.84 | 0.49–6.97 | 0.370 | – | | |
| Heart failure | 0.72 | 0.18–2.92 | 0.644 | – | | |
| Dyslipidemia | 0.86 | 0.55–1.34 | 0.495 | – | | |
| Obesity | 0.36 | 0.04–3.23 | 0.360 | – | | |
| None | 2.22 | 0.62–8.02 | 0.222 | – | | |
| Other comorbidities | 1.61 | 0.83–3.13 | 0.163 | – | | |
| **Complications** | | | | | | |
| Retinopathy | 1.15 | 0.76–1.76 | 0.507 | – | | |
| Nephropathy | 1.17 | 0.45–3.03 | 0.749 | – | | |
| Neuropathy | 1.04 | 0.66–1.63 | 0.859 | | | |
| Peripheral arterial disease | 0.97 | 0.16–5.85 | 0.970 | – | | |
| Foot ulcer or amputation | 1.16 | 0.31–4.41 | 0.824 | – | | |
| Stroke | 0.51 | 0.16–1.65 | 0.262 | – | | |
| **QOL z-score<−1SD** | 0.88 | 0.48–1.57 | 0.664 | – | | |
| **DSM score<−1SD** | 1.03 | 0.57–1.86 | 0.914 | – | | |

HbA1c: Glycated hemoglobin. cOR: Crude Odds Ratio. CI: Confidence Interval. aOR: Adjusted Odds Ratio. GH₵: Ghana Cedis. DM: Diabetes mellitus. QOL: Quality of life. DSM: Diabetes self-management

care versus tertiary care settings. The definition of glycemic control whether based on FBG or glycated hemoglobin also varies across the studies.

Our study showed that not being on insulin was associated with good glycemic control. Ninety-five percent of patients in our study had Type 2 DM, and as such are initiated on oral DM medications only to require insulin as the disease advances and there is depletion of beta islet cell mass. This finds continuity with a volume of previous works that indicated that instead of inherent poor glycemic control, insulin use reflects the seriousness of the condition of the patient [40,41]. One of the drawbacks of DM management is inertia in the initiation of insulin on the part of clinicians as well as patients [42]. Thus delayed insulin initiation would result in worsening of glycemic control. Our experience seems to conform to insulin use being associated with poor glycemic control by indication. In sharp contrast, other studies have also found that good glycemic control was associated with insulin use [30,38].

The physical health and social relationships domains of QOL had the lowest scores however, the overall QOL of patients in our study population was good in comparison with similar studies in Sub-Saharan Africa [11,12]. A study done in Nigeria showed that diabetic patients generally had lower QOL scores because of poor access to health care and other socioeconomic problems [43]. Income levels also played a significant role in determining QOL. In this case, better QOL was noted if respondents earned between two and ten times the minimum wage (GH₵ 541–2700, that is, USD 46–230) compared to household incomes with less than twice the minimum wage. It appears that, these respondents had sufficient resources to cope with their condition. This finding is consistent with past studies conducted in Africa, yielding the conclusion that at precariously low income levels, QOL might generally be adversely affected due to psychological distress [44,45]. It also emerged that recent hospitalization conferred a significant risk for poor QOL, which agrees with a study in China pointing out that acute health episodes hold negative implications for chronic disease management and general well-being alike [46]. DSM has been shown to be associated with good QOL in patients with DM. This was corroborated in our study as well as previous studies as discussed by a systematic review of randomized controlled trials [47]. Patients

**Table 6. Predictors of poor quality of life among patients with diabetes mellitus.**

| Variables | cOR | 95%CI | p-value | aOR | 95%CI | p-value |
|---|---|---|---|---|---|---|
| **Age in years** | 1.00 | 0.98–1.03 | 0.969 | 1.03 | 1.00–1.07 | 0.087 |
| **Gender, Male** | 1.60 | 0.86–2.99 | 0.137 | 1.10 | 0.46–2.66 | 0.830 |
| **Residence** | | | | | | |
| Rural | 1 | | | 1 | | |
| Peri-urban | 0.38 | 0.13–1.09 | 0.072 | 0.25 | 0.06–0.87 | 0.052 |
| Urban | 0.15 | 0.06–0.41 | <0.001 | 0.24 | 0.06–0.87 | 0.030 |
| **Marital status** | | | | | | |
| Single | 1 | | | – | | |
| Married | 0.49 | 0.19–1.25 | 0.137 | – | | |
| Separated | 0.30 | 0.03–2.74 | 0.285 | – | | |
| Divorced | 2.88 | 0.77–10.77 | 0.117 | – | | |
| Widowed | 0.52 | 0.18–1.47 | 0.217 | – | | |
| **Educational level** | | | | | | |
| Basic | 1 | | | – | | |
| Secondary | 0.56 | 0.29–1.10 | 0.090 | – | | |
| Tertiary | 0.70 | 0.30–1.62 | 0.409 | – | | |
| **Employment status** | | | | | | |
| Employed | 1 | | | – | | |
| Unemployed | 1.23 | 0.67–2.24 | 0.502 | – | | |
| **Household income (GH₵)** | | | | | | |
| <540 | 1 | | | 1 | | |
| 541–2700 | 0.34 | 0.18–0.63 | 0.001 | 0.28 | 0.13–0.64 | 0.002 |
| >2700 | 0.90 | 0.31–2.60 | 0.847 | 1 | 0.31–5.38 | 0.720 |
| **Type 2 DM** | 0.33 | 0.12–0.93 | 0.036 | 0.29 | 0.05–1.58 | 0.151 |
| **Disease duration ≥10years** | 1.03 | 0.56–1.90 | 0.917 | – | | |
| **Duration in clinic** | | | | | | |
| <1year | 1 | | | 1 | | |
| 1-3years | 0.47 | 0.16–1.40 | 0.176 | 0.36 | 0.08–1.63 | 0.185 |
| >3years | 0.41 | 0.20–0.87 | 0.021 | 0.77 | 0.26–2.30 | 0.640 |
| **Recent hospitalization** | 4.19 | 2.00–8.78 | <0.001 | 4.58 | 1.58–13.26 | 0.005 |
| **Family history of DM** | 0.53 | 0.30–0.94 | 0.031 | 0.82 | 0.38–1.78 | 0.618 |
| **Alcohol use** | | | | | | |
| Nil | 1 | | | – | | |
| Current | 0.54 | 0.12–2.38 | 0.412 | – | | |
| Past | 0.98 | 0.48–2.02 | 0.961 | – | | |
| **Smoking history** | | | | | | |
| Nil | 1 | | | – | | |
| Current | 0 | | | – | | |
| Past | 0.38 | 0.05–2.98 | 0.359 | – | | |
| **Number of DM medications used** | | | | | | |
| None or mono | 1 | | | – | | |
| Dual or multi | 0.94 | 0.46–1.93 | 0.865 | – | | |
| **DM medications used** | | | | | | |
| Insulin | 1 | | | – | | |
| No insulin | 0.57 | 0.32–1.02 | 0.057 | – | | |

*(Continued)*

**Table 6.** (Continued)

| Variables | cOR | 95%CI | p-value | aOR | 95%CI | p-value |
|---|---|---|---|---|---|---|
| **Comorbidities** | | | | | | |
| Hypertension | 1.04 | 0.49–2.22 | 0.912 | – | | |
| Chronic kidney disease | 1.61 | 0.32–7.94 | 0.561 | – | | |
| Heart failure | 1.61 | 0.32–7.94 | 0.561 | – | | |
| Dyslipidemia | 0.64 | 0.33–1.23 | 0.181 | – | | |
| Obesity | 1.39 | 0.15–12.71 | 0.769 | – | | |
| None | 1.40 | 0.29–6.78 | 0.675 | – | | |
| Other comorbidities | 2.11 | 0.96–4.64 | 0.061 | – | | |
| **Complications** | | | | | | |
| Retinopathy | 0.95 | 0.53–1.68 | 0.848 | – | | |
| Nephropathy | 3.90 | 1.44–10.55 | 0.007 | 2.15 | 0.59–7.82 | 0.244 |
| Neuropathy | 1.49 | 0.78–2.86 | 0.228 | – | | |
| Peripheral arterial disease | 3.80 | 0.62–23.28 | 0.149 | – | | |
| Foot ulcer or amputation | 0.69 | 0.08–5.61 | 0.726 | – | | |
| Stroke | 2.95 | 0.97–8.99 | 0.057 | – | | |
| **FBG<7** | 0.88 | 0.49–1.58 | 0.664 | – | | |
| **HbA1c<7** | 1.04 | 0.49–2.19 | 0.92 | – | | |
| **DSM score<−1SD** | 16.6 | 8.46–33.60 | <0.001 | 18.30 | 7.98–44.5 | <0.001 |

cOR: Crude Odds Ratio. CI: Confidence Interval. aOR: Adjusted Odds Ratio. GH₵: Ghana Cedis. DM: Diabetes mellitus. FBG: Fasting blood Glucose. HbA1c: Glycated hemoglobin. DSM: Diabetes self-management

who practice good DSM are expected to have better glucose control and therefore relatively reduced complications leading to better QOL. This is consistent with another systematic review which showed that linkage to good drug programing DSM improved health service and good psychological health outcomes [48].

Subsequent analysis demonstrated that poor DSM was not only associated with poor global QOL but was also associated with worse physical health, psychological health, social relationships and environment domains of QOL independently. Focusing on the physical health and social relationships domains that had the lowest scores, the following observations were made. Advancing age and other comorbidities such as cancers, chronic lung diseases and neuropsychiatric conditions contributed to multi-morbidity, and were associated with poor physical health. This is not surprising as these factors affect activities of daily living, mobility, pain and discomfort, and work capacity [32,49]. Facets of physical health domain should especially be assessed in the clinic among patients with advanced age and multi-morbidity. Also, male gender, past users of alcohol and patients with nephropathy had higher odds of poor social relationships. Males are more likely not to ask for help or accept social support from friends and family compared with females [50]. They are also more likely to not be satisfied with their sex life especially if there is sexual dysfunction. Internal components of sex like physical pleasure is most important for men, while satisfaction with interpersonal components of sex like emotional connection may be most important to women [51]. Availability of social support and satisfaction with sex life should be assessed in the clinic especially among males to improve their social health domain of QOL. Alcohol use on the other hand is known to be associated with social problems [52] while patients with chronic kidney disease have generally poor quality of life particularly related to poor social support [53,54].

Interventions for further improvement of DSM in our setting is promising because, overall, our population demonstrated better DSM compared to similar studies in Ethiopia and Myanmar [16,18]. Healthcare use domain of DSM had an almost perfect score meaning that participants kept regular appointments and reported health needs promptly. Our results imply an important

**Table 7. Predictors of poor diabetes self-managment among patients with diabetes mellitus.**

| Variables | cOR | 95%CI | p-value | aOR | 95%CI | p-value |
|---|---|---|---|---|---|---|
| **Age in years** | 1 | 0.97–1.02 | 0.857 | 1.01 | 0.98–1.05 | 0.400 |
| **Gender, Male** | 1.39 | 0.71–2.60 | 0.319 | 1.27 | 0.56–2.81 | 0.600 |
| **Residence** | | | | | | |
| Rural | 1 | | | 1 | | |
| Peri-urban | 0.83 | 0.29–2.61 | 0.733 | 1.08 | 0.32–3.98 | 0.900 |
| Urban | 0.19 | 0.07–0.60 | 0.003 | 0.30 | 0.09–1.04 | 0.047 |
| **Marital status** | | | | | | |
| Single | 1 | | | 1 | | |
| Married | 0.89 | 0.32–3.21 | 0.846 | 1.18 | 0.34–4.98 | 0.800 |
| Separated | 0.59 | 0.03–4.59 | 0.654 | 1.37 | 0.06–13.50 | 0.800 |
| Divorced | 5.69 | 1.37–26.90 | 0.020 | 6.92 | 1.24–43.20 | 0.031 |
| Widowed | 1.34 | 0.44–5.01 | 0.634 | 2.25 | 0.54–11.30 | 0.300 |
| **Educational level** | | | | | | |
| Basic | 1 | | | – | | |
| Secondary | 0.68 | 0.35–1.28 | 0.237 | – | | |
| Tertiary | 0.43 | 0.14–1.07 | 0.096 | – | | |
| **Employment status** | | | | | | |
| Employed | 1 | | | – | | |
| Unemployed | 1.15 | 0.63–2.08 | 0.639 | – | | |
| **Household income (GH₵)** | | | | | | |
| <540 | 1 | | | – | | |
| 541-2700 | 0.70 | 0.38–1.27 | 0.240 | – | | |
| >2700 | 0.42 | 0.06–1.55 | 0.259 | – | | |
| **Type 2 DM** | 0.59 | 0.20–2.13 | 0.362 | – | | |
| **Disease duration ≥10years** | 0.73 | 0.40–1.34 | 0.294 | – | | |
| **Duration in clinic** | | | | | | |
| <1year | 1 | | | 1 | | |
| 1-3years | 0.90 | 0.35–2.34 | 0.836 | 1.11 | 0.35–3.52 | 0.900 |
| >3years | 0.28 | 0.13–0.61 | <0.001 | 0.32 | 0.13–0.84 | 0.018 |
| **Recent hospitalization** | 2.41 | 1.05–5.20 | 0.030 | 1.85 | 0.62–5.13 | 0.300 |
| **Family history of DM** | 0.49 | 0.27–0.88 | 0.017 | 0.63 | 0.32–1.26 | 0.200 |
| **Alcohol use** | | | | | | |
| Nil | 1 | | | – | | |
| Current | 1.72 | 0.54–4.64 | 0.313 | – | | |
| Past | 0.85 | 0.37–1.78 | 0.679 | – | | |
| **Smoking history** | | | | | | |
| Nil | 1 | | | – | | |
| Current | 0 | | | – | | |
| Past | 0.88 | 0.14–3.33 | 0.873 | – | | |
| **Number of DM medications used** | | | | | | |
| None or mono | 1 | | | – | | |
| Dual or multi | 1.19 | 0.57–2.72 | 0.662 | – | | |
| **DM medications used** | | | | | | |
| Insulin | 1 | | | – | | |
| No insulin | 1.23 | 0.68–2.21 | 0.485 | – | | |

*(Continued)*

**Table 7.** (Continued)

| Variables | cOR | 95%CI | p-value | aOR | 95%CI | p-value |
|---|---|---|---|---|---|---|
| **Comorbidities** | | | | | | |
| Hypertension | 1.10 | 0.49–2.80 | 0.828 | – | | |
| Chronic kidney disease | 1.68 | 0.25–7.19 | 0.525 | – | | |
| Heart failure | 0.72 | 0.04–4.04 | 0.758 | – | | |
| Dyslipidemia | 0.68 | 0.34–1.28 | 0.249 | – | | |
| Obesity | 9.15 | 1.48–70.8 | 0.017 | 5.11 | 0.58–51.30 | 0.130 |
| None | 4.10 | 1.02–14.90 | 0.034 | 5.53 | 1.18–23.4 | 0.021 |
| Other comorbidities | 1.06 | 0.42–2.67 | 0.902 | – | | |
| **Complications** | | | | | | |
| Retinopathy | 0.72 | 0.40–1.30 | 0.279 | – | | |
| Nephropathy | 3.14 | 1.05–8.50 | 0.029 | 3.00 | 0.87–9.40 | 0.066 |
| Neuropathy | 1.41 | 0.75–2.79 | 0.307 | – | | |
| Peripheral arterial disease | 3.97 | 0.51–24.50 | 0.136 | – | | |
| Foot ulcer or amputation | 1.68 | 0.25–7.19 | 0.525 | – | | |
| Stroke | 2.20 | 0.59–6.71 | 0.193 | – | | |
| **FBG<7** | 0.97 | 0.54–1.77 | 0.914 | – | | |
| **HbA1c<7** | 0.72 | 0.35–1.50 | 0.379 | – | | |

cOR: Crude Odds Ratio. CI: Confidence Interval. aOR: Adjusted Odds Ratio. GH₵: Ghana Cedis. DM: Diabetes mellitus. FBG: Fasting blood Glucose. HbA1c: Glycated hemoglobin.

role of social support in DSM. Lack of needed support in elderly people who live without the other partner, to help manage their illness tended to have poor DSM practices. This finding is corroborated by a study in Ethiopia where being divorced was associated with poor self-care practices [55]. In the present study, being divorced was associated with about fourteen times odds of having poor dietary control. Patients without comorbidities had poor DSM, and particularly poor dietary control scores. It may be said that these patients seldom have encounters with the health system, hence fewer opportunities for direct education and DSM interventions. Those having multi-morbidity may be more motivated to undertake self-management and have more access to a number of knowledge and support sources. This aligns with outcomes from a study that demonstrated that comorbidities are often a driver of how active patients proceed to manage health issues [56]. Obesity on the other hand was associated with poor dietary control. A cross-sectional study from Ghana showed the inverse association between dietary carbohydrate quality index and obesity [57]. Dietary control is a crucial component of DSM as it directly influences blood glucose levels. The critical role of dieticians in the management of patients with DM cannot be overemphasized.

Patients with nephropathy and stroke had poorer physical activity mainly due to the physical limitation of their complications. Our study also showed that, those who had attended the clinic for over three years demonstrated the best overall DSM and physical activity scores. This is probably influenced by the fact that patients who have attended clinic for a longer duration, have been exposed to multiple education sessions as organized daily by the DM unit in the hospital as well as have received counselling from doctors and dieticians which have led to empowering these patients to practice DSM. On the other hand, this finding raises serious concerns about the efficiency of current unstructured DSM interventions. At face value, it means that the current unstructured interventions may take up to three years for significant improvement in DSM to be realized. It may thus take structured, culturally appropriate and probably more intensive DSM interventions to achieve better outcomes in a short time.

The glucose management domain of DSM had the worst scores and the only protective factor was urban residence. The glucose management domain encompasses medication adherence and self-monitoring of blood glucose [33].

# PLOS One

**Table 8. Predictors of poor individual QOL domains.**

| Characteristic | Bivariate | | | Multivariable | | |
|---|---|---|---|---|---|---|
| | cOR[1] | 95% CI[1] | p-value | aOR[1] | 95% CI[1] | p-value |
| **Physical health** | | | | | | |
| Age in years | 1.03 | 0.99, 1.07 | 0.112 | 1.10 | 1.05, 1.15 | <0.001 |
| Urban residence | 0.11 | 0.04, 0.41 | <0.001 | 0.11 | 0.03, 0.49 | 0.003 |
| Type 2 DM | 0.26 | 0.09, 0.99 | 0.028 | 0.04 | 0.01, 0.33 | 0.002 |
| Recent hospitalization | 5.14 | 2.05, 12.20 | <0.001 | 11.00 | 2.75, 45.20 | <0.001 |
| Other comorbidities | 3.13 | 1.16, 7.63 | 0.016 | 6.08 | 1.76, 20.40 | 0.003 |
| DSM score<−1SD | 7.51 | 3.32, 17.10 | <0.001 | 6.62 | 2.50, 18.20 | <0.001 |
| **Psychological health** | | | | | | |
| Married | 0.43 | 0.19, 1.00 | 0.043 | 0.33 | 0.12, 0.92 | 0.030 |
| Unemployed | 1.83 | 1.09, 3.15 | 0.026 | 3.28 | 1.61, 7.05 | 0.002 |
| Other comorbidities | 3.25 | 1.63, 6.45 | <0.001 | 3.84 | 1.54, 9.36 | 0.003 |
| Neuropathy | 2.75 | 1.53, 5.22 | 0.001 | 3.92 | 1.93, 8.61 | <0.001 |
| DSM score<−1SD | 15.8 | 8.13, 32.3 | <0.001 | 18.70 | 8.47, 44.5 | <0.001 |
| **Social relationships** | | | | | | |
| Male | 3.32 | 2.02, 5.50 | <0.001 | 2.99 | 1.63, 5.52 | <0.001 |
| Income (GH₵) 541–2700 | 0.37 | 0.23, 0.60 | <0.001 | 0.31 | 0.18, 0.52 | <0.001 |
| Past alcohol use | 2.74 | 1.59, 4.72 | <0.001 | 2.16 | 1.15, 4.04 | 0.016 |
| Nephropathy | 4.84 | 1.83, 14.20 | 0.002 | 4.14 | 1.46, 12.90 | 0.009 |
| DSM score<−1SD | 3.58 | 1.97, 6.57 | <0.001 | 3.48 | 1.80, 6.80 | <0.001 |
| **Environment** | | | | | | |
| Primary level of education | 3.99 | 1.47, 14.00 | 0.014 | 5.79 | 1.75, 24.50 | 0.008 |
| Income (GH₵) 541–2700 | 0.28 | 0.14, 0.53 | <0.001 | 0.31 | 0.14, 0.68 | 0.004 |
| DSM score<−1SD | 15.70 | 7.91, 31.80 | <0.001 | 16.20 | 6.83, 41.00 | <0.001 |

[1]cOR = Odds Ratio, CI = Confidence Interval, aOR: Adjusted Odds Ratio.

GH₵: Ghana Cedis. DM: Diabetes mellitus. QOL: Quality of life. DSM: Diabetes self-management

Barriers to medication adherence are myriad and have been encountered in several previous studies [16–19,30,37]. Self-monitoring of blood glucose on the other hand is strongly linked to access to glucometers and test strips as well as technical ability to perform the test and document appropriately [28,58]. These factors may explain why patients in urban centers have better glucose management. Medication adherence should thus be assessed at clinic visits and buttressed with sustained adherence counselling as part of DSM interventions. Also patients from rural areas should be supported to be able to access glucometers, either personally or from nearby health facilities, and encouraged to get help with documentation of their blood glucose results.

Some limitations should be noted. Causal inferences cannot be made aside the observed associations as this was a cross-sectional study. Glycemic control was assessed using both fasting blood glucose and glycated hemoglobin independently, however some participants did not have glycated hemoglobin done due to financial constraints. The generalisability of the study to the Ghanaian population is limited as the study was conducted in a single tertiary hospital. However, the KATH receives referrals from all over Ghana especially the middle and northern regions of Ghana.

## Conclusion

This study underscores a holistic approach to the real world management of DM as experienced by both healthcare providers and patients. 4 out of 10 patients attending diabetes clinic are well controlled. Not being on insulin is independently

**Table 9. Predictors of poor individual DSM domains scores.**

| Characteristic | Bivariate | | | Multivariable | | |
|---|---|---|---|---|---|---|
| | cOR[1] | 95% CI[1] | p-value | aOR[1] | 95% CI[1] | p-value |
| **Glucose management** | | | | | | |
| Urban residence | 0.31 | 0.11, 0.84 | 0.017 | 0.33 | 0.12, 0.92 | 0.029 |
| **Dietary control** | | | | | | |
| Divorced | 9.33 | 1.80, 72.00 | 0.013 | 14.30 | 2.44, 124.00 | 0.006 |
| Obesity | 7.23 | 1.17, 55.80 | 0.032 | 7.77 | 1.05, 71.60 | 0.045 |
| Nephropathy | 3.18 | 1.13, 8.44 | 0.022 | 3.40 | 1.15, 9.42 | 0.020 |
| No comorbidities | 7.55 | 2.09, 30.3 | 0.002 | 8.53 | 2.22, 36.70 | 0.002 |
| **Physical activity** | | | | | | |
| Urban residence | 0.14 | 0.05, 0.37 | <0.001 | 0.11 | 0.04, 0.33 | <0.001 |
| >3years clinic attendance | 0.43 | 0.21, 0.93 | 0.025 | 0.37 | 0.16, 0.92 | 0.029 |
| Nephropathy | 3.63 | 1.28, 9.67 | 0.011 | 5.36 | 1.63, 17.4 | 0.005 |
| Neuropathy | 2.31 | 1.19, 4.86 | 0.019 | 3.46 | 1.62, 8.14 | 0.002 |
| Stroke | 3.76 | 1.21, 10.90 | 0.016 | 5.20 | 1.55, 16.60 | 0.006 |
| **Healthcare utilization** | | | | | | |
| Recent hospitalization | 5.45 | 2.16, 13.00 | <0.001 | 5.54 | 1.99, 14.90 | <0.001 |
| Peripheral arterial disease | 8.80 | 1.12, 55.50 | 0.020 | 11.60 | 1.43, 76.20 | 0.010 |

[1]cOR: Crude Odds Ratio. CI: Confidence Interval. aOR: Adjusted Odds Ratio.DM: Diabetes mellitus. QOL: Quality of life. DSM: Diabetes self-management

associated with good glycemic control. Urban residence is positively associated with QOL while poor DSM and recent hospitalization are adversely associated with QOL. Patients who are divorced have worse DSM while patients who have attended the diabetes clinic for more than three years have better DSM. Prevention of acute hospitalizations and promotion of good self-management among patients with diabetes can improve their quality of life. Also, good social support and unstructured DSM interventions experienced over at least three years of clinic attendance can improve patients' self-management. Future studies can be targeted at implementing sustained, intensive, culturally appropriate, structured interventions aimed at improving glycemic control, DSM and ultimately quality of life of patients with diabetes mellitus.

## Supporting information

**S1 File. Diabetes study dataset.**
(XLS)

**S2 File. STROBE Statement—Checklist of items that should be included in reports of cross-sectional studies.**
(DOCX)

## Acknowledgments

We would like to acknowledge all the staff at the DM clinic at Komfo Anokye Teaching Hospital for their contribution toward this study.

## Author contributions

**Conceptualization:** Kwadwo Faka Gyan, Enoch Agyenim-Boateng, Kojo Awotwi Hutton-Mensah, Priscilla Abrafi Opare-Addo, Solomon Gyabaah, Elliot Koranteng Tannor.

**Methodology:** Kwadwo Faka Gyan, Enoch Agyenim-Boateng, Kojo Awotwi Hutton-Mensah, Priscilla Abrafi Opare-Addo, Solomon Gyabaah, Emmanuel Ofori, Osei Yaw Asamoah, Mohammed Najeeb Naabo, Michael Asiedu Owiredu, Elliot Koranteng Tannor.

**Project administration:** Kwadwo Faka Gyan, Enoch Agyenim-Boateng.

**Supervision:** Kwadwo Faka Gyan, Enoch Agyenim-Boateng, Kojo Awotwi Hutton-Mensah, Priscilla Abrafi Opare-Addo, Solomon Gyabaah, Elliot Koranteng Tannor.

**Validation:** Kwadwo Faka Gyan, Enoch Agyenim-Boateng, Kojo Awotwi Hutton-Mensah, Priscilla Abrafi Opare-Addo, Solomon Gyabaah, Emmanuel Ofori, Osei Yaw Asamoah, Mohammed Najeeb Naabo, Michael Asiedu Owiredu, Elliot Koranteng Tannor.

**Writing – original draft:** Kwadwo Faka Gyan, Kojo Awotwi Hutton-Mensah, Priscilla Abrafi Opare-Addo, Solomon Gyabaah, Elliot Koranteng Tannor.

**Writing – review & editing:** Kwadwo Faka Gyan, Enoch Agyenim-Boateng, Kojo Awotwi Hutton-Mensah, Priscilla Abrafi Opare-Addo, Solomon Gyabaah, Emmanuel Ofori, Osei Yaw Asamoah, Mohammed Najeeb Naabo, Michael Asiedu Owiredu, Elliot Koranteng Tannor.

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
