## [Decision Letter · Decision Letter 0]

PONE-D-24-58978Predictors of glycemic control, quality of life and diabetes self-management of patients with diabetes mellitus at a tertiary hospital in GhanaPLOS ONE

Dear Dr. Gyan,

Thank you for submitting your manuscript to PLOS ONE. After careful consideration, we feel that it has merit but does not fully meet PLOS ONE’s publication criteria as it currently stands. Therefore, we invite you to submit a revised version of the manuscript that addresses the points raised during the review process.

ACADEMIC EDITOR: Please make the suggested changes before resubmitting.

We look forward to receiving your revised manuscript.

Kind regards,

Yee Gary Ang, MBBS MPH

Academic Editor

PLOS ONE

Journal Requirements:

Reviewers' comments:

Reviewer's Responses to Questions

Comments to the Author

1. Is the manuscript technically sound, and do the data support the conclusions?

Reviewer #1: Yes

Reviewer #2: Yes

Reviewer #3: Partly

2. Has the statistical analysis been performed appropriately and rigorously? 

Reviewer #1: Yes

Reviewer #2: Yes

Reviewer #3: Yes

3. Have the authors made all data underlying the findings in their manuscript fully available?

Reviewer #1: Yes

Reviewer #2: Yes

Reviewer #3: Yes

4. Is the manuscript presented in an intelligible fashion and written in standard English?

Reviewer #1: Yes

Reviewer #2: Yes

Reviewer #3: Yes

5. Review Comments to the Author

Reviewer #1: Line 87: delete the 4th word :and"

Line 96: Start a new sentence with the 5th word :However"

Line 147: Delete the 2nd word "and"

Line 157: Change the 8th word "include" to "included"

Line 168: Indicate which type of statistical software you used for the analysis here

Line 279: Start as a new sentence

Line 287: Start as a new sentence

Lines 304-306: What reasons can you assign to your finding that insulin use is associated with poor glycaemic control in the study population?

Reviewer #2: The study underscores a holistic approach to the real world management of DM as experienced

by both healthcare providers and patients. The study finding add to scientific knowledge, The manuscript was well written and structurally organized.

However the authors need to check table 1, the frequency for the female was missing (line 188, p8)

Reviewer #3: This work looked at a very salient area in diabetic management. However, I have some comments to help improve the quality of the manuscript. I am attaching my comments since it exceeds the word limit.

6. PLOS authors have the option to publish the peer review history of their article (what does this mean?). If published, this will include your full peer review and any attached files.

Do you want your identity to be public for this peer review? For information about this choice, including consent withdrawal, please see our Privacy Policy.

Reviewer #1: Yes: Prof. Baafuor Opoku

Reviewer #2: No

Reviewer #3: No

---

## [Author Response · Author response to Decision Letter 1]

29 Mar 2025

Response to reviewers

The authors appreciate the extensive review done and are happy to incorporate the corrections, recommendations and suggestions to improve this manuscript.

Reviewer #1:

Line 87: delete the 4th word :and"

Response: Thanks for the correction. The change has been appropriately effected in the manuscript. (Line 88)

Line 96: Start a new sentence with the 5th word :However"

Response: Thanks for the correction. The change has been appropriately effected in the manuscript. (Line 97)

Line 147: Delete the 2nd word "and"

Response: Thanks for the correction. The change has been appropriately effected in the manuscript. (Line 153)

Line 157: Change the 8th word "include" to "included"

Response: Thanks for the correction. The change has been appropriately effected in the manuscript. (Line 163)

Line 168: Indicate which type of statistical software you used for the analysis here

Response: Thanks for the correction. The change has been appropriately effected in the manuscript. (Line 174)

Line 279: Start as a new sentence

Response: Thanks for the correction. The change has been appropriately effected in the manuscript. (Line 305)

Line 287: Start as a new sentence

Response: Thanks for the correction. The change has been appropriately effected in the manuscript. (Line 313)

Lines 304-306: What reasons can you assign to your finding that insulin use is associated with poor glycaemic control in the study population?

Response: The reasons as proposed in the manuscript (Lines 332-345) include:

-Most participants (95% in the current study being Type 2 DM) require insulin only when their glycemic control is already bad. This is a form of association by indication.

-Another reason may be inertia on the part of doctors and patients to initiate insulin in patients who require insulin due to poor glycemic control. Thus delayed insulin initiation would result in worsening of glycemic control.

-Barriers to insulin use which may contribute to poor glycemic control were not fully elucidated in the current study. These include hypoglycemia and fear of hypoglycemia, weight gain, needle anxiety, insufficient education, lack of access to insulin and poor insulin storage. The need for glucose monitoring and insulin dose adjustment may also be a challenge as this present study shows the weakest domain of DSM to be glucose management.

Reviewer #2:

The study underscores a holistic approach to the real world management of DM as experienced

by both healthcare providers and patients. The study finding add to scientific knowledge, The manuscript was well written and structurally organized.

However the authors need to check table 1, the frequency for the female was missing (line 188, p8)

Response: Thanks for the correction. The change has been appropriately effected in the manuscript. (Line 196)

Reviewer #3:

This work looked at a very salient area in diabetic management. However, I have some comments to help improve the quality of the manuscript. I am attaching my comments since it exceeds the word limit.

General notice:

- In the formulation of the sentences, it will be noticed that the ‘period’ (full stop) which marks the end of a sentence is most often placed before the in-text references thus distorting the end of a previous statement and beginning of the next statement.

- Check on this a correct appropriately for general conformance.

- Example Line 64.

Response: Thanks for the correction. The changes have been appropriately effected in the entire manuscript.

Ln 69-70 � The opening sentence ‘several studies have demonstrated that compare to the general population…’, needs to be revised. It can be understood that authors would want to paraphrase to avoid plagiarism but the sentence restructuring in its current form does not offer clarity.

Response: Thanks for the correction. The sentence has been restructured for clarity. (Line 70-72)

Ln 84 � Consider revising the use of the pronoun ‘his’ as the case group comprises male and female.

Sentence should be more generic to cover both genders.

Response: Thanks for the correction. The change has been appropriately effected in the manuscript. (Line 85)

Ln 87 � Revise the use of ‘care’ as it stands vaguely in the context of the sentence.

What does ‘its’ as used refer to?

Check the sentence and revise appropriately.

Response: Thanks for the correction. The sentence has been restructured for clarity. (Line 88)

Ln 88 � In what context is ‘overall mean scores’ used? Example, overall mean score for what?

Revise the sentence

Response: Thanks for the correction. The sentence has been revised appropriately. (Line 89)

Ln 102 � The abbreviation for DSM has been presented as DMS

Response: Thanks for the correction. The change has been appropriately effected in the manuscript. (Line 104)

Ln 147 � Negatively should read ‘negative’

Response: Thanks for the correction. The change has been appropriately effected in the manuscript. (Line 153)

Ln 155 � Delete ‘the’ preceding ‘patient’s’

Response: Thanks for the correction. The change has been appropriately effected in the manuscript. (Line 161-162)

Ln 162 � BMI as used must be defined on first use before subsequently presented in abbreviation format.

Response: Thanks for the correction. The change has been appropriately effected in the manuscript. (Line 169)

Ln 167 � Provide the manufacturer details for the REDCap software

Response: Thanks. The manufacturer details have been provided. (Line 173-174)



Ln 168 � The first letter of ‘Characteristics’ must not be capitalized

Response: Thanks for the correction. The change has been appropriately effected in the manuscript. (Line 175)

Ln 187 � The period should be placed after (Table 1) and not before it as done.

Response: Thanks for the correction. The change has been appropriately effected in the manuscript. (Line 195)

Table 1

The inclusion of the study site in the heading of the table is not relevant.

Response: Thanks for the correction. The change has been appropriately effected in the manuscript. (Line 196)

Male should be placed in parenthesis

Response: Thanks for the recommendation. Both male and female have been provided in the table and formatted uniformly. (Line 196)

A total number of 360 has been given as part of the table heading but it was realized that the total number presented for HbA1c was 215

Presenting 215 as the number for HbA1c proves that is a shortfall but no reason has been given in the results section to support this finding.

The percentage estimated from the 215 was used for discussions without recourse to the shortfall.

Response: Thanks for this salient observation. The fact that some members of the study population did not have evaluable HbA1c was discussed as part of the limitations of this study with specific reason adduced. (Line 423-425)

The reason for the short fall is further given in the results section and subsequent discussions have been put in context. (Lines 42-44, 203-204, 214-216, 241, 252 and 316-323)

For characteristic which are presented with / (NHIS/Private Insurance), what do they mean? Does the slash present two items as alternatives? Does the item after the slash represent the point of focus? Authors should check this and correctly.

Examples: No response/Unknown; Foot ulcer/amputation

Response: Thanks for the recommendation. The slash represents alternatives. This has been corrected appropriately in all the tables. (Lines 196, 252, 258, 264 and 270)

Number of medications

What does 0-1 mean in medication delivery? Between 0-1 what did authors count?

Consider using the categories: none, 1 (mono), 2 (dual) and 2+ for multi

Response: Thanks for the recommendation. This has been corrected appropriately in all the tables. (Lines 196, 252, 258, 264 and 270)

Ln 190 � Check the spelling for mellitus

Response: Thanks for the correction. The change has been appropriately effected in the manuscript. (Line 198)

Ln 192-193 � SGLT-2i and GLP1 were not administered to patients and no record of these were presented on table 1. Delete as part of the footnote.

Why are these presented under the footnote?

Authors should refrain from throwing in variables which hitherto were not assessed just because they intend to include such unexplored data in the discussion.

Response: Thanks for the observation and we apologize for the misrepresentation. The authors investigated specifically the use of these medications among participants and recorded zero usage. It is unfortunate that the data presented on the Table 1 had SGLT-2i and GLP-1 analogues omitted because the frequency and percentage was 0. This omission has been appropriately corrected in Table 1. (Line 196)

Ln 202 � There is no break point for the previous sentence and as such it follows through into the ensuing sentence.

Response: Thanks for the correction. The change has been appropriately effected in the manuscript. (Line 214)

Ln 203 � The percentage estimated for HbA1c (44.7%) is being used without recourse to the fact that it was estimated over 215 and not 360.

215 represents approximately 60% of the studied population for whom there was evaluable HbA1c data and this must always be factored into the narrative as it has its place in subsequently analysis and clinical extrapolations.

Response: Thanks for this salient observation. The fact that some members of the study population did not have evaluable HbA1c was discussed as part of the limitations of this study with specific reason adduced. (Line 423-425)

The reason for the short fall is further given in the results section and subsequent discussions have been put in context. (Lines 42-44, 203-204, 214-216, 241, 252 and 316-323)

Ln 209-210 � There was no need for this statement once again because no information of these drugs was assessed. It can therefore not be added to the presentation just by inclusion to lay foundation to be used for an angle of the discussion as was done in Line 287

Delete

Response: Thanks for the observation and we apologize sincerely for the misrepresentation. The authors investigated specifically the use of these medications among participants and recorded zero usage. It is unfortunate that the data presented on the Table 1 had SGLT-2i and GLP-1 analogues omitted because the frequency and percentage was 0. This omission has been appropriately corrected in Table 1. (Line 196)

Ln 224 � should read logistic and not ‘logistics’

Response: Thanks for the correction. The change has been appropriately effected in the manuscript. (Line 239)

Line 231 � there should be a space after hospitalization and the parenthesis

Response: Thanks for the correction. The change has been appropriately effected in the manuscript. (Line 247)

Ln 237 � HbA1c was evaluable for 215 patients.

145 patients did not have HbA1c results.

What is the measure of good glycaemic control in these 145 patients?

Can FBS be relied upon solely as a guide to good glycaemic control in these patients.

Response: Thanks for this salient observation. Mean FBG in mmol/L of participants with HbA1c (n=215) is 9.1±4.9 and of participants without HbA1c (n=145) is 8.8 ±4.7 (p=0.541) (Line 203-204)

In the present study, there was no statistical difference between the mean FBG of participants with HbA1c and those without HbA1c hence FBG could be relied upon as a guide to good glycemic control in these patients. (Line 320-322)

Line 290 � the sentence ‘good glycaemic control was observed in 44.7% of patients’ is not a true picture of the findings.

The 44.7% was estimated from 215 and not 360

215 represents approximately 60% of the participants.

The 44.7% estimated only reflects information for 60% of the participants and this must be given attention.

In reality, authors missed the point that 40% of the patients are unable to access HbA1c testing.

In the absence of HbA1c results, what will be the best measure of glycaemic control in these patients?

Response: Thanks for this salient observation. Subsequent discussions have been put in context. (Lines 42-44, 203-204, 214-216, 241, 252 and 316-323)

Alternatives to assessing glycemic control when HbA1c is not accessible may include using FBG, post prandial glucose and self-monitoring of blood glucose. (Line 322-323)

Ln 312-314 � Requires revision.

Response: Thanks for the recommendation. The discussion has been appropriately revised to read: “Income levels also played a significant role in determining QOL. In this case, better QOL was noted if respondents earned between two and ten times the minimum wage (GH₵ 541 to 2700, that is, USD 46 to 230) compared to household incomes less than twice the minimum wage. It appears that, these respondents had sufficient resources to cope with their condition. This finding is consistent with past studies conducted in Africa, yielding the conclusion that at precariously low income levels, QOL might generally be adversely affected due to psychological distress [46, 47].” (Lines 350-355)

Discussion

Very little discussion was made about the domains of QOL and diabetes self-management score. This takes away immensely from the focus of the study since emphasis was laid on the QOL and DSM.

Emphasis was laid on the independent study variables and their relationship with the dependent variables as assessed yet, not discussion on the domains and scores.

The study did not attempt to evaluate how the covariates influence the QOL domains and DSM scores.

The discussion would have to be revised to include information on which aspects and section of the DSM and QOL questionnaire should be paid attention to in their routine use for long term patient management.

Response: Thanks for the recommendation. The methodology and discussion has been appropriately expanded to include the domains of QOL and DSM scores. (Lines 141-146, 182-183, 346, 365-383, 386-387, 391-393, 398-400, 410-421 and 433)

Further analysis has been done to evaluate how the covariates influence the QOL domains and DSM scores. (Lines 182-183, 274-293, 295 and 297)

Critical aspects of the QOL and DSM questionnaire for routine long term patient management have been highlighted. (Lines 141-146, 372-373, 379-380 and 417-421)

Conclusion

Not suitable for the scope of work done because the analysis fails to explore a relationship between the covariates and the domains of QOL and scores of DSM. This would have thrown much light into areas of the QOL and DSM questionnaires where medical personnel would have to pay much attention during educational sessions for the patients.

As regards QOL, social relationship domain scored the least. The question now arises that what specific steps must be taken to improve this domain? This was not addressed in the discussion but attempts were made to conclude on such albeit not much analysis was conducted on such.

In DSM, glucose management score was the least among the other scores. This observation was not raised for effective discussion to enable readers appreciate the steps that medical personnel will explore to raise the glucose management score to high values and ultimately ensure good quality of life.

The conclusion as it stands does not address any of the concerns raised by way of arguments in the background section.

Response: Thanks for the recommendation. The methodology and discussion has been appropriately expanded to include the domains of QOL and DSM scores. (Lines 141-146, 182-183, 346, 365-383, 386-387, 391-393, 398-400, 410-421 and 433)

Further analysis has been done to evaluate how the covariates influence the QOL domains and DSM scores. (Lines 182-183, 274-293, 295 and 297)

Further discussions have particularly focused on social relationships domain of QOL and glucose management domain of DSM exploring avenues for improvement. (Lines 373-383 and 410-421)

The conclusion therefore lies within the scope of the work done and addresses arguments raised in the background.

Kindly note the following changes to the reference list.

Ten new references have been added during the revision process:

36. Ketema EB, Kibret KT. Correlation of

---

## [Decision Letter · Decision Letter 1]

PONE-D-24-58978R1Predictors of glycemic control, quality of life and diabetes self-management of patients with diabetes mellitus at a tertiary hospital in GhanaPLOS ONE

Dear Dr. Gyan,

Thank you for submitting your manuscript to PLOS ONE. After careful consideration, we feel that it has merit but does not fully meet PLOS ONE’s publication criteria as it currently stands. Therefore, we invite you to submit a revised version of the manuscript that addresses the points raised during the review process.

**ACADEMIC EDITOR: One reviewer has accepted but another reviewer still has unresolved comments. **

We look forward to receiving your revised manuscript.

Kind regards,

Yee Gary Ang, MBBS MPH

Academic Editor

PLOS ONE

Journal Requirements:

Reviewers' comments:

Reviewer's Responses to Questions

**Comments to the Author**

1. If the authors have adequately addressed your comments raised in a previous round of review and you feel that this manuscript is now acceptable for publication, you may indicate that here to bypass the “Comments to the Author” section, enter your conflict of interest statement in the “Confidential to Editor” section, and submit your "Accept" recommendation.

Reviewer #2: All comments have been addressed

Reviewer #3: (No Response)

2. Is the manuscript technically sound, and do the data support the conclusions?

Reviewer #2: Yes

Reviewer #3: Partly

3. Has the statistical analysis been performed appropriately and rigorously? 

Reviewer #2: Yes

Reviewer #3: Yes

4. Have the authors made all data underlying the findings in their manuscript fully available?

Reviewer #2: Yes

Reviewer #3: Yes

5. Is the manuscript presented in an intelligible fashion and written in standard English?

Reviewer #2: Yes

Reviewer #3: Yes

6. Review Comments to the Author

Reviewer #2: The authors have duly addressed all the commen. The manuscript is tecnically sound and data supports the conclusion.

The manuscript was written in standard English

Reviewer #3: The authors tried to respond to the queries I raised in the first revision. However, there still remain some few comments that needs to be answered. This has been attached as a separate document.

7. PLOS authors have the option to publish the peer review history of their article (what does this mean?). If published, this will include your full peer review and any attached files.

Reviewer #2: No

Reviewer #3: **Yes: **Christian Obirikorang

---

## [Author Response · Author response to Decision Letter 2]

3 Jun 2025

Response to reviewers

Thank you very much for the extensive review done to improve this manuscript. The authors are happy to incorporate the corrections, recommendations and suggestions.

Reviewer #3:

The authors tried to respond to the queries I raised in the first revision. However, there still remain some few comments that needs to be answered. This has been attached as a separate document.

Line Comment

39 Multivariable logistics should read multivariable logistic

Response: The change has been effected. (Line 39)

62 Mentions the overall prevalence of DM as 6.46% with no year reference and this was compared with 0.2% in 1964. Which year relative to 1964 is being evaluated?

Response: The statement has been revised to include the year being referenced. “according to a systematic review and meta-analysis by Asamoah-Boaheng et al in 2019”. (Line 64)

131 The ‘q’ term for the Fisher formula was not defined.

Response: All the terms for the Fisher’s formula have been defined. “q = (1- p)”. (Line 135)

172 It indicates that poor DSM was defined as DSM score <6 making reference to Schmitt et al., (1996).

A critical evaluation of the scoring guide indicates that the use of a cut-off score of <6 to detect persons with suboptimal or non-adherent self-management behaviours is not recommended. The statement is as refers below:

NOT RECOMMENDED: To detect persons with suboptimal or non-adherent self-management behaviours, a cut-off score of < 6 on the total scale has been utilized in some studies – This is not recommended! This criterion most likely originates from an abstract publication (see ref. no. 4 above) reporting on a study in which 226 German PWD were categorised according to their DSMQ total scores using median split. (“the median split yielded a cut-off score of ≈ 6

on the DSMQ scale ranging from 0 to 10.”) By design, this score split the sample into two halves, reflecting that a cut-off score of < 6 would identify about half of the sample with suboptimal self-management behaviour, thus not being suitable for detecting clinical cases.

Was the choice of a cut-off score <6 arrived at based on a median split for the cohort of patients being evaluated?

Response: The choice of cut-off score <6 was not based on a median split for the current cohort of patients being evaluated but was based on the abstract referenced in your comment. The investigators deliberated on the options of using -1 SD as cut-off as was used for QOL in the present study. However, the previously documented cut-off in literature of <6 was settled on without keen observation of its limitations. The authors have therefore established a cut-off using the current population distribution characteristic at -1 SD from the mean. This new analysis output has been used in the relevant tables of results. (Line 174)

230 In Line 158, authors indicated a percentage transformation of scale scores ranging from 0 to 100.

1. How was a score <6 chosen as a cut-off for poor DSM on a 0 to 100 scale?

The DSM scores stated in this section ranged from 0 to 10. No score exceeded 10 thus defying the transformation score of 0 to 100 stated by authors.

Response: The actual scale scores range from 0 to 10. The authors have corrected the error in Line 158 which should have read “Sums of item scores are calculated to give scale scores and then converted into a scale that ranges from 0 to 10 (raw score/theoretical maximum score *10).” (Lines 159-161)

236 The sum score is 10 and not 100 as expected from authors statement in Line 158

Response: The authors have corrected the error in Line 158 which should have read “Sums of item scores are calculated to give scale scores and then converted into a scale that ranges from 0 to 10 (raw score/theoretical maximum score *10).” (Lines 159-161)

240-241 The statement reads: ‘patients on only oral DM medications (aOR 2.14; 95% CI 1.19-3.88, p=0.012) were more likely to have good glycemic control compared to those on insulin when HbA1c (n=215) was used as a measure of glycemic control (Table 4a).

1. From table 4a, under DM medications used, the covariates evaluated were ‘Dual or Multi’ and ‘no insulin’. Authors evaluated ‘no insulin’ but in the statement were presenting information on insulin use. This is contradictory and must be re-evaluated.

2. The aOR of 2.14 was an adjusted OR for ‘no insulin’ in the table but this was presented as an output for oral medications which had no results outcome in the adjusted analysis column.

3. The analysis from the table shows that patients who are not on insulin had good glycaemic control compared to others.

Response: The authors recognize the incompleteness of the labelling of variables in the (Tables 4a to 6) under DM medications as against the actual analysis done which leads to the misinterpretation of the results. These analysis outputs should have been reported separately, expanded under two different headings i.e “Number of DM medications used” and “DM medications used” (ref Table 1). Under Number of DM medications used “Dual or multi” was compared to “None or mono” while under DM medications used “No insulin” was compared to “Insulin”. This correction has been made in all the relevant tables to give an accurate representation of the analysis performed. (Lines 254, 260, 266 and 272)

Also, in the abstract, results and subsequent discussion, the description of “not being on insulin” has been appropriately used instead of the description “only oral medications”. (Lines 42, 242, 306, 333 and 433)

242-243 The statement reads: ‘FBG as a measure of glycemic control yielded similar results (aOR 1.82; 95% CI 1.12-2.96,p=0.017)’

1. This statement was made drawing inference from the HbA1c statement in Line 240. It is worthy to note that the interpretation error made in Line 240 likewise runs through the assertion for this statement.

2. From the table Dual/Multi and No Insulin as covariates had outputs under the univariate and adjusted models.

3. From the output, no insulin is associated with good glycaemic control as measured by FBG in the univariate and adjusted models and the opposite for Dual/Multi medications.

4. It cannot therefore be the case or suggestion that oral medications were associated with good glycaemic control based on FBG because the analysis does not support that assertion.

Response: Same as above. This correction has been made in all the relevant tables to give an accurate representation of the analysis performed, and in the subsequent discussions. (Lines 254, 260, 266 and 272) and (Lines 42, 242, 306, 333 and 433)

248 Period is placed before the parenthesis.

Response: This has been corrected. (Line 250)

295 Table 7

Based on the foregoing arguments on the use of 6 as a score cut-off, further analysis based on score cut-off ≈6 requires re-evaluation.

Response: The authors have recognized this and as such subsequent analysis have been based on an established cut-off using the current population distribution characteristic at -1 SD from the mean. The new analysis output has been used in all the relevant tables of results.

(Lines 254, 260, 266 and 272)

302-303 The statement reads: ‘on only oral DM medications to be associated with good glycemic control as compared with patients on insulin’

1. This assertion is not supported by the evidence from the analysis shown on Tables 4a and 4b. The tables evaluated ‘no insulin’ and there was no indication from the analysis that oral medications were associated with good glycaemic control.

2. Take note, no analysis from the results section supports oral medication over insulin.

Response: Same as above. This correction has been made in all the relevant tables to give an accurate representation of the analysis performed, and in the subsequent discussions. (Lines 254, 260, 266 and 272) and (Lines 42, 242, 306, 333 and 433)

303 Statements like, ‘Socio-economic factors such as urban residence and household income were protective of poor QOL’, can be expressed in a better way in the discussion than the persistent use of qualifiers such as ‘poor QOL’ in the analysis.

What does protective of poor QOL mean in this scenario?

Response: This statement and other such persistent use of qualifiers have been revised in the discussion and relevant sections of the manuscript. (Line 307)

304 ‘Good overall DSM was also associated with good QOL’ statements like this can be expressed better to give better meaning than the format of applying qualifiers.

Response: This statement has been revised as recommended. The analysis has been done with poor DSM (< -1 SD) to be the covariate with the output to suggest its association with worse QOL appropriately. This correction has been made in all the tables and discussed appropriately without the need for too many qualifiers. (Line 308 ) and (Lines 254, 260, 266, 272, 299 and 301)

314 Statement was made about SGLT-2 inhibitors and GLP-1 agonists and how beneficial they are in diabetic patient populations. The fact remains that none of the patients evaluated were on SGLT-2 inhibitors and GLP-1 agonists and there was no way the benefits of these drugs could be assessed in this study.

This statement is therefore not relevant.

Given the fact that no patient were on these drugs and no analysis on benefit was conducted, the statement on gap in access cannot be supported in this study.

Response: The two statements have been omitted.

321 ‘FBG could be relied upon as a guide to good…’

1. FBG is already in use as a diagnostic criterion for diabetes care and management.

2. The fact that authors could did not observed any statistical difference in mean FBG for patients with HbA1c and those without HbA1c is not scientific and clinical proof enough to suggest and propose FBG as a guide to good glycaemic control.

Response: The authors have omitted this statement.

323 Self-monitoring of blood glucose was stated as an alternative to assessing glycaemic control when HbA1c is no accessible.

Fact is, self-monitoring of blood glucose will still be limited to FBG estimation and as such does not provide any varied alternative to FBG which was listed in addition to post prandial glucose.

The section needs revision

Response: The authors have revised this section. Self-monitoring of blood glucose provides a dynamic blood glucose profile rather than a single snapshot of blood glucose. It is therefore useful in resource-poor settings where ambulatory continuous glucose monitoring systems are unavailable. (Line 324,324)

328 The differences in glycemic control across studies and countries may be due to differences in treatment strategies, access to specialized care, DSM education and socio-cultural factors

This too general of a statement for which critical analysis was not conducted in this study. One would want to know the specific item lines associated with treatment strategies, access to specialized care and socio-cultural factors mediating differences in glycaemic control.

Response: This section has been revised and focuses on only the studies referenced. “The differences in glycemic control across these studies may be due to the study designs: single-center versus multi-center and primary care versus tertiary care settings. The definition of glycemic control whether based on FBG or glycated hemoglobin also varies across the studies.” (Lines 329-332)

331 The statement reads: ‘Our study showed that being on only oral DM medications was associated with good glycemic control’

This assertion is not supported by the analysis as indicated in comments regarding medication above. This assertion must be revised based on corrections and deductions drawn from the analysis

Response: Same as above. This correction has been made in all the relevant tables to give an accurate representation of the analysis performed, and in the subsequent discussions. (Lines 254, 260, 266 and 272) and (Lines 42, 242, 306, 333 and 433)

338 The statement reads: Our experience seems to conform to insulin use being associated with poor glycemic control by indication.

As the analysis will show, insulin use was not assessed in this study. What was assessed was ‘no insulin’ and the interpretation on this must be well reviewed by authors.

Response: Same as above. This correction has been made in all the relevant tables to give an accurate representation of the analysis performed, and in the subsequent discussions. (Lines 254, 260, 266 and 272) and (Lines 42, 242, 306, 333 and 433)

340-341 The statement reads: Barriers to insulin use which may contribute to poor glycemic control were not fully elucidated in the current study

The study never assessed barriers to insulin use and as such cannot suggest or ascribe reasons to that which was not evaluated in the discussion.

Response: This statement has been omitted.

411 The conclusion as it stands does not fully address the true findings from the study and the potential lessons and implications for diabetic patient management. Consider revising.

Response: The conclusion has been revised to fully address the true findings from the study. Lessons and implications for management of patients with diabetes have also been summarized. (Lines 50-54 and 430-442)

Kindly note the following changes to the reference list.

Reference 43 and 44 have been removed during the revision process.

43. Bayked EM, Kahissay MH, Workneh BD. Barriers and facilitators to insulin treatment: a phenomenological inquiry. J Pharm Policy Pract [Internet]. 2022 Dec 1 [cited 2025 Mar 12];15(1):1–11. Available from: https://joppp.biomedcentral.com/articles/10.1186/s40545-022-00441-z

44. Chen R, Aamir AH, Feroz Amin M, Bunnag P, Chan SP, Guo L, et al. Barriers to the Use of Insulin Therapy and Potential Solutions: A Narrative Review of Perspectives from the Asia–Pacific Region. Diabetes Ther [Internet]. 2024 Jun 1 [cited 2025 Mar 12];15(6):1261–77. Available from: https://link.springer.com/article/10.1007/s13300-024-01568-9

One new reference 57 has been added.

57. Suara SB, Siassi F, Saaka M, Rahimi Foroshani A, Sotoudeh G. Association between Carbohydrate Quality Index and general and abdominal obesity in women: a cross-sectional study from Ghana. BMJ Open. 2019;9(12):e033038. doi:10.1136/BMJOPEN-2019-033038

---

## [Decision Letter · Decision Letter 2]

Predictors of glycemic control, quality of life and diabetes self-management of patients with diabetes mellitus at a tertiary hospital in Ghana

PONE-D-24-58978R2

Dear Dr. Gyan,

We’re pleased to inform you that your manuscript has been judged scientifically suitable for publication and will be formally accepted for publication once it meets all outstanding technical requirements.

Kind regards,

Yee Gary Ang, MBBS MPH

Academic Editor

PLOS ONE

Additional Editor Comments (optional):

Reviewers' comments:

Reviewer's Responses to Questions

**Comments to the Author**

1. If the authors have adequately addressed your comments raised in a previous round of review and you feel that this manuscript is now acceptable for publication, you may indicate that here to bypass the “Comments to the Author” section, enter your conflict of interest statement in the “Confidential to Editor” section, and submit your "Accept" recommendation.

Reviewer #3: All comments have been addressed

2. Is the manuscript technically sound, and do the data support the conclusions?

Reviewer #3: Yes

3. Has the statistical analysis been performed appropriately and rigorously? 

Reviewer #3: Yes

4. Have the authors made all data underlying the findings in their manuscript fully available?

Reviewer #3: Yes

5. Is the manuscript presented in an intelligible fashion and written in standard English?

Reviewer #3: Yes

6. Review Comments to the Author

Reviewer #3: With regards to the current form of the manuscript, all my concerns raised have been addressed. The manuscript can therefore be accepted

7. PLOS authors have the option to publish the peer review history of their article (what does this mean?). If published, this will include your full peer review and any attached files.

Reviewer #3: **Yes: **Christian Obirikorang

---

## [Editor Report · Acceptance letter]

PONE-D-24-58978R2

PLOS ONE

Dear Dr. Gyan,

I'm pleased to inform you that your manuscript has been deemed suitable for publication in PLOS ONE. Congratulations! Your manuscript is now being handed over to our production team.

Kind regards,

on behalf of

Dr. Yee Gary Ang

Academic Editor

PLOS ONE